

# Molecular evolution and expression divergence of three key Met biosynthetic genes in plants: *CGS*, *HMT* and *MMT*

Man Zhao[1], Wenyi Wang[1], Lei Wei[1], Peng Chen[1], Fengjie Yuan[2], Zhao Wang[1] and Xiangxian Ying[1]

[1] College of Biotechnology and Bioengineering, Zhejiang University of Technology, Hangzhou, China
[2] Institute of Crop Science, Zhejiang Academy of Agricultural Sciences, Hangzhou, China

## ABSTRACT

Methionine (Met) is an essential sulfur-containing amino acid in animals. Cereal and legume crops with limiting levels of Met represent the major food and feed sources for animals. In plants, cystathionine gamma-synthase (CGS), methionine methyltransferase (MMT) and homocysteine methyltransferase (HMT) are committing enzymes synergistically synthesizing Met through the aspartate (Asp) family pathway and the S-methylmethionine (SMM) cycle. The biological functions of *CGS*, *MMT* and *HMT* genes have been respectively studied, whereas their evolution patterns and their contribution to the evolution of Met biosynthetic pathway in plants are unknown. In the present study, to reveal their evolution patterns and contribution, the evolutionary relationship of *CGS*, *MMT* and *HMT* gene families were reconstructed. The results showed that *MMT*s began in the ancestor of the land plants and kept conserved during evolution, while the *CGS*s and *HMT*s had diverged. The *CGS* genes were divided into two branches in the angiosperms, Class 1 and Class 2, of which Class 2 only contained the grasses. However, the *HMT* genes diverged into Class 1 and Class 2 in all of the seed plants. Further, the gene structure analysis revealed that the *CGS*s, *MMT*s and *HMT*s were relatively conserved except for the *CGS*s in Class 2. According to the expression of *CGS*, *HMT* and *MMT* genes in soybeans, as well as in the database of soybean, rice and *Arabidopsis*, the expression patterns of the *MMT*s were shown to be consistently higher in leaves than in seeds. However, the expression of *CGS*s and *HMT*s had diverged, either expressed higher in leaves or seeds, or showing fluctuated expression. Additionally, the functions of *HMT* genes had diverged into the repair of S-adenosylmethionine and SMM catabolism during the evolution. The results indicated that the *CGS* and *HMT* genes have experienced partial subfunctionalization. Finally, given the evolution and expression of the *CGS*, *HMT* and *MMT* gene families, we built the evolutionary model of the Met biosynthetic pathways in plants. The model proposed that the Asp family pathway existed in all the plant lineages, while the SMM cycle began in the ancestor of land plants and then began to diverge in the ancestor of seed plants. The model suggested that the evolution of Met biosynthetic pathway is basically consistent with that of plants, which might be vital to the growth and development of different botanical lineages during evolution.

Corresponding authors
Man Zhao, mzhao@zjut.edu.cn
Xiangxian Ying, yingxx@zjut.edu.cn

## INTRODUCTION

Methionine (Met) is an essential amino acid which is mainly obtained from human and animal foods. Met plays important functions, not only as a protein component and in the initiation of mRNA translation, but also in indirectly regulating various metabolic processes through its main catabolic product, S-adenosylmethionine (SAM, AdoMet) (*Galili & Amir, 2013*; *Roje, 2006*; *Sauter et al., 2013*). Despite its important functions, Met could only be synthesized in plants and microorganisms (*Galili & Amir, 2013*).

The biosynthetic pathway of Met has been widely studied, and can be synthesized by most bacteria, fungi and plants. In bacteria and fungi, Met is mainly synthesized in four steps from homoserine: homoserine—O-succinylhomoserine—cystathionine—homocysterine—Met, which are catalyzed by homoserine O-succinyltransferase, cystathionine gama-synthase, cystathionine beta-lyase and Met synthase, respectively (*Ferla & Patrick, 2014*). However, in *Corynebacterium glutamicum*, O-succinylhomoserine is replaced by O-acetylhomoserine. There is also another pathway from O-acetylhomoserine direct to Met catalyzed by O-acetylhomoserine sulfhydrylase (*Bolten et al., 2010*; *Willke, 2014*).

In plants, Met can be synthesized through the aspartate (Asp) family pathway as well as the S-methylmethionine (SMM) cycle. In the Asp family pathway, homoserine is firstly converted into O-phosphohomoserine (OPH) by homoserine kinase. Then, the condensation reaction of OPH with cysteine is catalyzed into cystathionine by cystathionine gamma-synthase (CGS), and cystathionine is hydrolyzed into homocysteine (Hcy) through cystathionine beta-lyase. Next, Met is synthesized de novo through Met synthase (*Datko, Giovanelli & Mudd, 1974*). As for the SMM cycle, methionine methyltransferase (MMT) uses Met, synthesized by the Asp family pathway, and SAM to form SMM, then SMM and Hcy are converted into two molecules of Met through the catalysis of homocysteine methyltransferase (HMT) (*Bourgis et al., 1999*; *Ranocha et al., 2001*; *Lee et al., 2008*; *Cohen et al., 2017a*).

Furthermore, the biosynthesis process of Met during the development of plants is revealed by genetic and biochemical experiments. First, Met is synthesized by the Asp family pathway in rosette leaves, in which it is converted into SMM by MMT; second, the SMM is translocated into reproductive tissues, such as siliques and seeds, and reconverted back into Met in the developing seeds by HMT (*Bourgis et al., 1999*; *Ranocha et al., 2001*; *Lee et al., 2008*; *Cohen et al., 2017b*; *Kocsis et al., 2003*). Additionally, the possible contribution of SMM to the stress effects was also proposed (*Cohen et al., 2017b*). Above all, CGS, HMT and MMT are essential enzymes in the synthesis of Met.

CGS is the mainly regulatory enzyme in the Asp family pathway. In *Arabidopsis*, when CGS was constitutively over-expressed, the soluble Met and SMM accumulated in specific stages, such as flowers, siliques, seedling tissues and roots of mature plants (*Kim et al., 2002*). In contrast, the repression of *CGSs* made the plants abnormal and produced partial

Met auxotrophy (*Kim & Leustek, 2000*). Interestingly, when the seed-specific repression of *CGS* was performed, more SMMs were transported from the leaves to reproductive organs, in which there were higher reconversion rates of SMM to Met, and more Met was accumulated in seeds (*Cohen et al., 2017b*). In addition, studies have reported that the expressions of *CGS* were in the negative feedback regulation of their products, Met or SAM, in wild-type *Arabidopsis* (*Kim & Leustek, 2000*; *Thompson et al., 1982*; *Ranocha et al., 2000*). Further, the MTO1 region in the first exon of *AtCGS* was proven to result in its negative feedback regulation (*Chiba et al., 1999*; *Ominato et al., 2002*). In *mto1* mutants of *AtCGS1*, both the enzyme levels and soluble Met levels were increased (*Chiba et al., 1999*). Afterward, the seed-specific expression of the feedback-insensitive form of *AtCGSs* in plants were also studied, but with different results (*Cohen et al., 2014*, *2016*, *2017b*; *Hanafy et al., 2013*; *Song et al., 2013*; *Matityahu et al., 2013*). For example, in *Arabidopsis*, soybean and tobacco, the sulfur-associated metabolism was altered and the soluble Met was significantly elevated in seeds. However, there was no Met increase in azuki bean (*Matityahu et al., 2013*). Therefore, the *CGS* gene family might have diverged in different organisms during evolution.

HMT and MMT are essential in the SMM cycle (*Cohen et al., 2017a*; *Zhao et al., 2018*). The evolution and expression of *HMT*s have been studied, as detailed in our previous research. Research found that *HMT*s have diverged into two clades in seed plants and that their expression also diverged. It has been proposed that the divergence of *HMTs* might be crucial to meeting the needs of plant development and growth (*Zhao et al., 2018*). As for MMT, it was only studied in *Arabidopsis* by catalyzing the synthesis of SMM from Met and AdoMet (*Ranocha et al., 2001*). Nevertheless, the systematic evolution patterns of the three key enzymes, CGSs, HMTs and MMTs, in plants, and how they contribute to the evolution of the Met biosynthesis pathway are unclear.

Soybean is an important economic crop, as it is a source of vegetable proteins in the human diet. In soybean seeds, major storage proteins consist of glycinin (11S) and conglycinin (7S), and 11S proteins account for approximately 30% (*Nielsen et al., 1989*; *Harada, Barker & Goldberg, 1989*). The sulfur-containing Met is an essential amino acid, the level of which often limits the nutritional value of crop plants (*Galili et al., 2005*). Therefore, considering the importance of the three enzymes CGS, MMT and HMT in the synthesis of Met and soybeans, this study comprehensively analyzed their evolutionary history, including their phylogenetic relationship and gene structures, and examined their selection pressures. Their expression profiles in soybeans were also widely analyzed. Taken together, this research is helpful for understanding the evolutionary history and functional divergence of the *CGS*, *MMT* and *HMT* gene families in plants, and might also provide an overall picture of the evolutionary and functional model of the Met biosynthetic pathway in plants.

# MATERIALS AND METHODS

## Phylogenetic analysis

The gene sequences in full genomes of plants were examined with genes in *A. thaliana* as query. The sequences followed the criteria: *E*-value < $1 \times$ e-05 in the BLASTN and

TBLASTN programs, and an amino acid identity above 40%, which were downloaded from the databases of Phytozome (http://www.phytozome.net/), congenie (http://congenie.org/) and NCBI (https://www.ncbi.nlm.nih.gov/). Altogether, 49 *CGS* sequences and 43 *MMT* sequences were obtained from the major plant lineages studied (Tables S1 and S2 in File S3). Multiple alignments of gene sequences were executed in the Clustal X v1.81 program with default parameters and alignments, optimized via manual adjustments using BioEdit v 7.0.9.0 (*Thompson et al., 1997*; *Hall, 1999*). Maximum likelihood (ML) and Neighbor-Joining (NJ) trees were reconstructed using PhyML online with the GTR + G + I model and MEGA6 software (*Guindon & Gascuel, 2003*; *Tamura et al., 2013*). The resultant trees were represented using MEGA 6. The phylogenetic tree of *HMT* genes has been shown in our previous study (*Zhao et al., 2018*).

## Analysis of gene structure

The intron and exon structures of *CGS* and *MMT* genes were analyzed according to their genome sequences and coding sequences. The length and numbers of introns and exons were shown in Table S3 in File S3. In addition, the conserved motifs in proteins were detected using the Multiple Em for Motif Elicitation (MEME) server (http://meme-suite.org/tools/meme) (*Bailey et al., 2009*). The server was run using the default values and choices. We conducted the search for 16 motifs in proteins arbitrarily. The motifs retrieved by MEME were reported according to their statistical significance, and the most statistically significant (low *E*-value) ones were shown first. The *E*-value of a motif is based on its log likelihood ratio, width, sites and the size of the set. The motifs of HMTs have been analyzed in our previous results (*Zhao et al., 2018*).

## Detection of selection pressures

To estimate the selection pressures in the gene families, the codeml program from the PAML v4.4 package, on the basis of codon sequence alignments, was performed (*Yang, 2007*). The likelihood ratio test is a general method for testing assumptions (model parameters) by comparing two competing hypotheses.

## Plant materials and growth conditions

The cultivated soybean "Chuandou 4" was grown at a farm in Fuyang (Hangzhou, China) during summer. Each materials of the leaves, stems, flowers and 2-, 3-, 4-, 5- and 6-week post-fertilization fruits for the gene expression study were harvested at the same time. The harvested tissues were immediately stored in liquid N2 and then stored at −80 °C for total RNA extraction using TRIzol reagent (Invitrogen, Carlsbad, CA, USA).

## Realtime RT-PCR analyses

Two micrograms of total RNA were used to synthesize the first strand cDNA using a ReverTra Ace qPCR RT Kit cDNA Synthesis Kit (TOYOBO). Quantitative RT-PCR (qRT-PCR) was conducted using a ChamQ TM SYBR qPCR Master Mix (Vazyme) in a CFX Connect Real-Time system (BIO-RAD, Hercules, CA, USA). ACTIN (Glyma.18G290800) was used as an internal control. Each experiment was performed

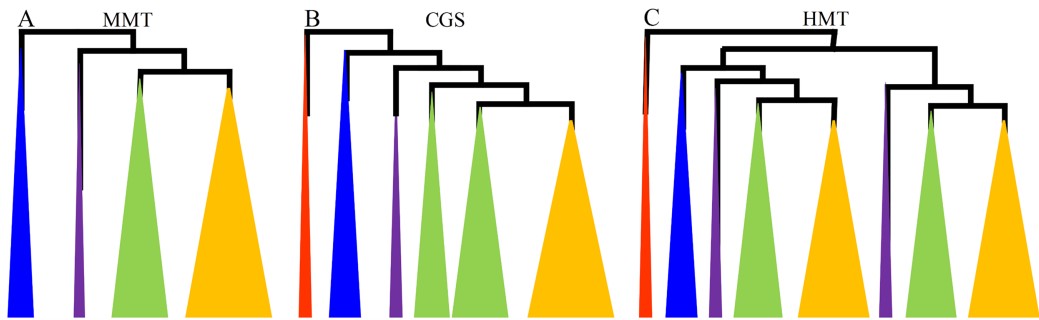

**Figure 1 The diagrams of the evolutionary relationships of *CGS*, *HMT* and *MMT* gene families.** (A–C) represent *MMT*, *CGS* and *HMT* gene family, respectively. The diagrams were based on their phylogenetic trees in Fig. S1 and *Zhao et al. (2018)*. The red, blue, purple, green and orange triangles represent algae, basal land plants, basal angiosperms and gymnosperms, monocots and dicots, respectively.

using three independent biological samples. PCR was performed in a 25.0 μL reaction mixture containing 5.0 μL Vazyme, 50 ng cDNA template, 0.4 μL of each primer (10.0 μM) and 3.2 μL of double distilled $H_2O$. The optimized operational procedure was performed as follows: 2 min at 95 °C (one cycle), 10 s at 95 °C, 30 s at 60 °C (40 cycles), 5 s at 65 °C and 5 s at 95 °C (one cycle for the melting curve analysis). The relative gene expression was evaluated as previously described (*Livak & Schmittgen, 2001*).

The expression of genes in different tissues was analyzed in the PLEXdb database (http://www.plexdb.org/index.php) (*Dash et al., 2012*).

### Promoters analysis

The promoter sequences (2,000 bp upstream of the transcription initiation site), *GmaHMTs*, *GmaCGSs* and *GmaMMTs*, were obtained from Phytozome. To identify the putative *cis*-acting regulatory elements, the promoter sequences of *GmaHMTs*, *GmaCGSs* and *GmaMMTs* were submitted to PlantCARE (http://bioinformatics.psb. ugent.be/webtools/plantcare/html/) (*Lescot et al., 2002*).

### Statistical analysis

In this study, standard deviations were calculated based on a minimum of three independent replicates. Comparative statistical analyses of groups were performed using Student's t test.

## RESULTS

### Identification and phylogenetic analysis of *CGS*, *MMT* and *HMT* genes

In this study, we reconstructed the phylogenetic trees of the *CGS* and *MMT* genes in plants to understand their evolutionary history. The genes from representative whole-genome plants lineages, which contained monocots, eudicots, basal angiosperms, gymnosperms, basal land plants and chlorophyta, were surveyed (Tables S1 and S2 in File S3). The phylogenetic trees were constructed using ML and NJ methods (File S1). Due to the similar topologies of ML and NJ trees, ML trees were shown with higher support values (Fig. 1; Figs. S1A–S1B in File S2).
First, the *CGS* genes were widely separated in plant lineages from algae to angiosperms. In total, 49 representative *CGS*s were used to reconstruct the phylogenetic tree, and the *CGS* genes in algae, as the outgroup, were located at the base. The *CGS* genes in gymnosperms and basal land plants had not diverged and were grouped in Class 3, while the *CGS*s had diverged into two classes in angiosperms, Class 1 and Class 2 (Fig. 1; Fig. S1A in File S2). In Class 1, all of the genes were contained in angiosperms, whereas, in Class 2, only the genes contained in grasses were present. The results indicated the *CGS* genes might have diverged asynchronously in angiosperms.

However, *MMT* genes were not found in algae, which appeared from the ancestor of land plants. Furthermore, *MMT* genes were relatively conserved, with only one or two copies, except for *PpaMMT*s with three copies. In total, 43 *MMT*s were surveyed. Phylogenetically, the evolutionary relationship of *MMT* genes with their species relationship was relatively consistent (Fig. 1; Fig. S1B in File S2). As for the *HMT* genes, we have reported that they existed in kinds of plant lineages, and they have diverged into two classes in all of the seed plants (Fig. 1) (*Zhao et al., 2018*). Therefore, the phylogenetic relationships of *CGS*, *MMT* and *HMT* genes showed that the *MMT*s were conserved, while the *HMT*s and *CGS*s had diverged in grasses and all of the seed plants, respectively. The results implied that these genes have experienced asynchronous divergence during their evolution.

## Intron–exon structures of *CGS*s, *MMT*s and *HMT*s

The divergence of genes is partly reflected in their structures. The intron–exon structures, as well as the number, length and position of introns and exons, in the *CGS*, *MMT* and *HMT* genes were analyzed (Table S3 in File S3).

The analysis of the number and position of introns and exons showed that most *CGS* genes in land plants contained an 11-exon and 10-intron pattern (Table S3A in File S3). For example, in Class 1, 26 (26/35 = 74.3%) genes maintained the pattern and the remaining nine had different degrees of intron gains or losses. In Class 3 all of the *CGS*s belonged to the 11-exon and 10-intron pattern, except for the unknown *PglCGS* and *PsiCGS*. However, in Class 2 the number of exons was less than 6 and half of them (5/10 = 50%) contained only two exons. The results showed that the divergence in exon–intron numbers have occurred between Class 2 and other classes in *CGS* genes. In addition, 76.2% (32/42 = 76.2%) of *MMT* genes contained a 12-exon and 11-intron pattern, while the remaining 23.8% (10/42 = 23.8%) experienced intron gain or loss events of different degrees (Table S3B in File S3). Generally, the exon–intron pattern of *MMT* genes was conserved during evolution, which was similar with *HMT* genes (70.67% *HMT*s had a 7-exon and 6-intron pattern) (*Zhao et al., 2018*).

Besides the number and position, the length of exons and introns was also considered in our study. In the three gene families, the corresponding lengths of exons were basically consistent, except for the *CGS* genes in Class 2, while the corresponding lengths of introns were various in all of the genes (Table S3 in File S3). Finally, the analysis of the exon–intron structures indicated that the structures of the *HMT* and *MMT* genes were conserved, while the intron–exon numbers in *CGS* genes diverged, especially in Class 2.

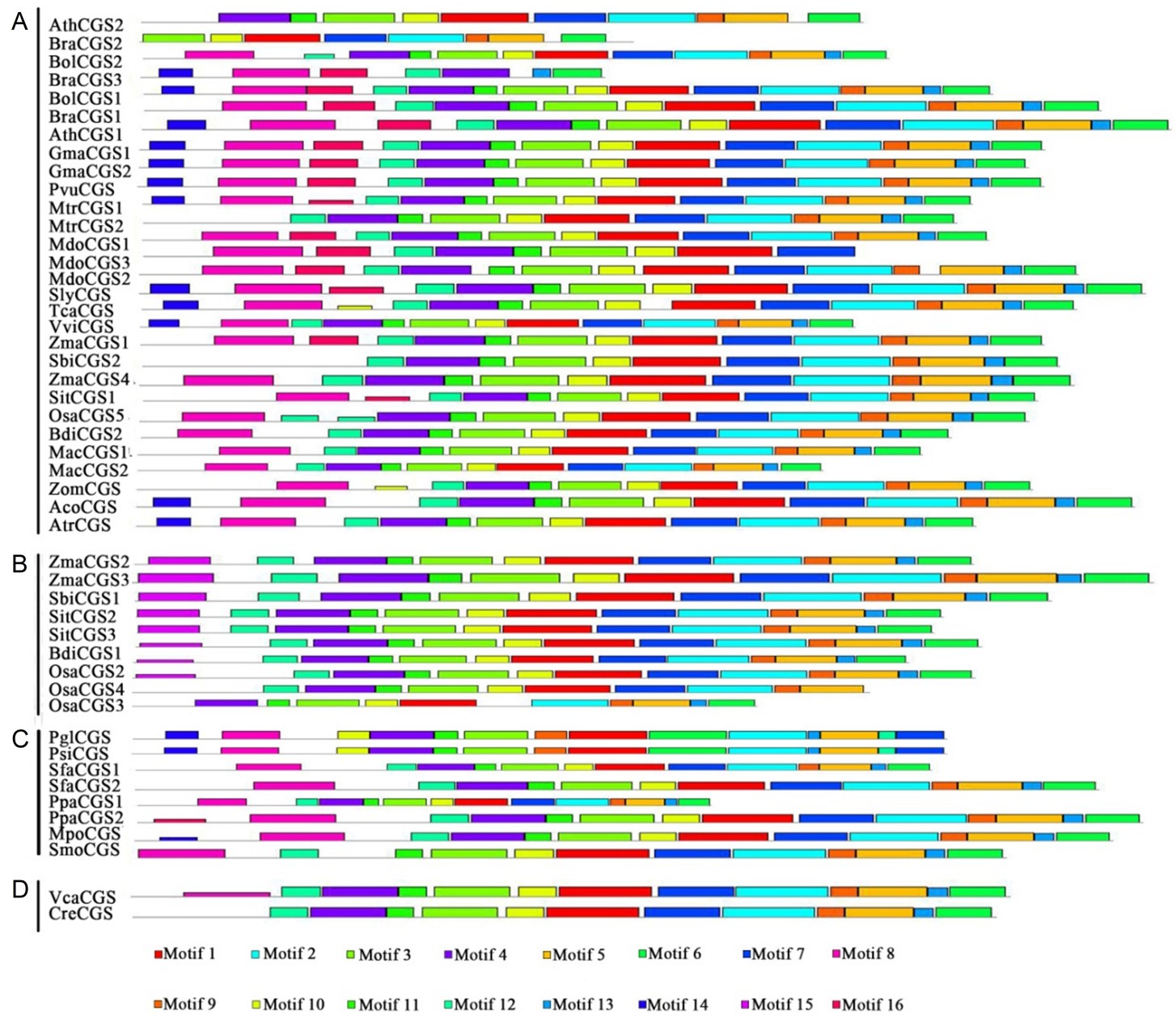

**Figure 2 Conserved motifs of CGS proteins identified on the MEME analysis across plants.** Each motif is represented by a colored box numbered on the bottom. (A–D) represent Class 1, Class 2, Class 3 and Outgroup, respectively. The amino acid sequences of these motifs are presented in Fig. S1 in File S2. The black lines represent unique sequences. The scale bar indicates number of amino acids. Names to the left indicate the clades to which the sequences belong according to Fig. S1 in File S2.

## Protein motifs analysis in MEME

Protein structures were analyzed to survey the conserved protein motifs of CGSs, MMTs and HMTs in MEME. In total, 16 motifs were identified and shown in CGSs (Fig. 2; Fig. S2A in File S2). Among them, 12 motifs (motif 1–motif 6 and motif 9–motif 14), located in the middle and *C*-terminal of the CGS proteins, were found in all CGS proteins (Fig. 2; Fig. S2A in File S2). However, the motifs in *N*-terminals, such as motif 7, 8, 15 and 16, were

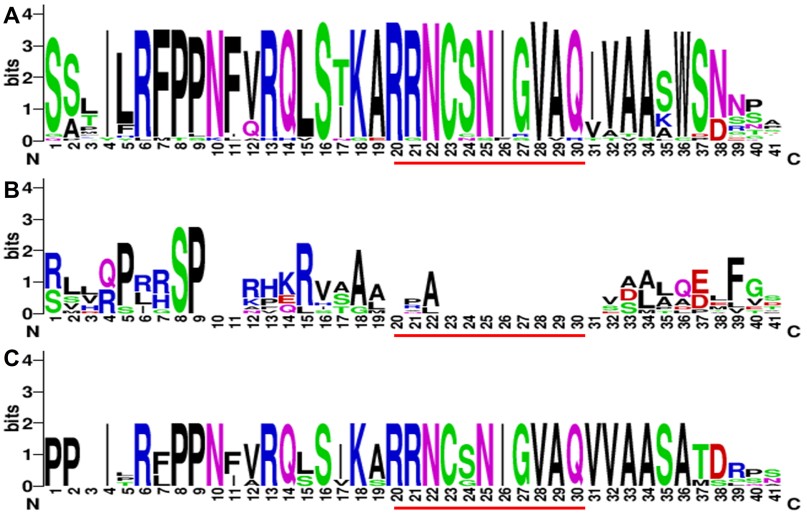

**Figure 3 The sequence composition of the conserved regions in Class 1, Class 2 and Class 3 in CGS family.** (A–C) represent the conserved region in Class 1, Class 2 and Class 3, respectively. The MTO1 region positions are marked by red lines. The height of each letter represents the probability of the letter at that position, and total height of the stack represents the information content of that position.

divided among classes. For instance, motifs 7, 8 and 16 were in Class 1 and Class 3, while motif 15 was in Class 2. The MTO1 region is essential for the negative feedback regulation of *CGS* genes, which is located in the *N*-terminals of CGS (marked in red lines in Fig. 3 and File S1). In this study, the MTO1 region was only found in motif 8. Hence, the CGSs in Class 2 had lost their MTO1 regions during evolution (Fig. 3). In addition, some *CGS*s in Class 1, such as *AthCGS2* and *BraCGS2*, lost their MTO1 regions. Furthermore, three key sites in the MTO1 region (R77, S81 and G84 in AtCGS1) were not detected in Class 2, AthCGS2 or BraCGS2. In view of the functions of the MTO1 region, the results indicated that the negative feedback regulation might have been lost in Class 2, *AthCGS2* and *BraCGS2*.

However, in MMT proteins, 16 motifs were totally consistent in all of the MMTs, except BdiMMT1 and RcoMMT, which indicated that the protein motifs of MMTs were conserved during evolution (Fig. 4; Fig. S2B in File S2). Similarly, the protein motifs of HMTs were also conserved (*Zhao et al., 2018*). Based on the results above, the divergence of protein motifs has occurred in CGS proteins, but not in HMTs and MMTs.

## The selection pressure of the *CGS*, *HMT* and *MMT* families

Selection pressure is used to identify the genes have undergone adaptive evolution. To analyze the selection pressure of the gene families, the $\omega$ values ($\omega = dN/dS$) were estimated, and the $\omega$ value was defined as the ratio of nonsynonymous and synonymous substitution. The results showed that the $\omega$ values of *CGS*s, *MMT*s and *HMT*s were 0.19, 0.17 and 0.16, respectively. The selection pressures showed that they were under stringent negative selection during evolution, and hence their functions were stringent conserved during evolution.

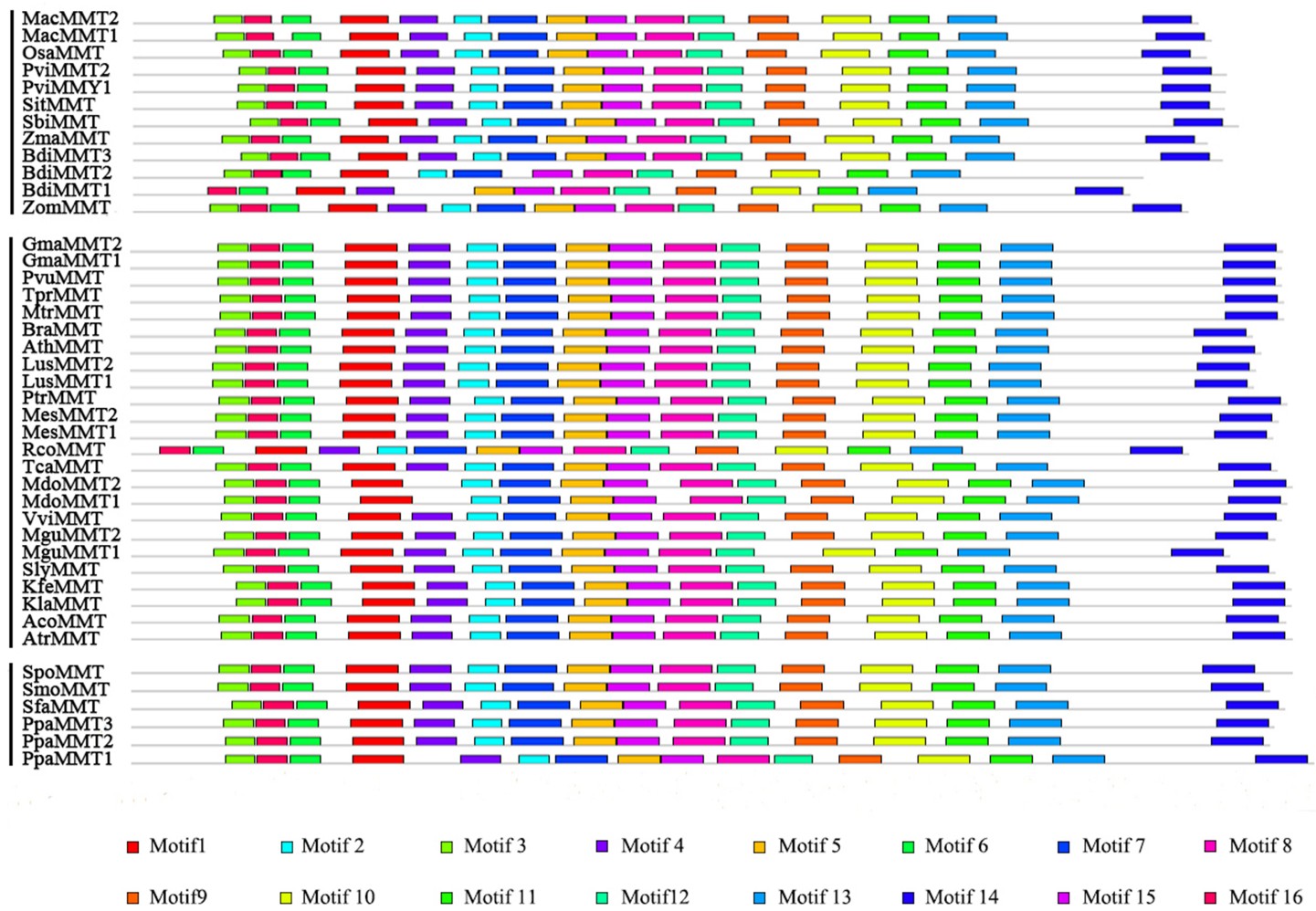

**Figure 4 Conserved motifs of MMT proteins identified on the MEME analysis across plants.** Each motif is represented by a colored box numbered on the bottom. The amino acid sequences of these motifs are presented in Fig. S1 in File S2. The black lines represent unique sequences. The scale bar indicates number of amino acids. Names to the left indicate the clades to which the sequences belong according to Fig. S1 in File S2.

## qRT-PCR analysis of *CGS*, *MMT* and *HMT* genes in soybean

The expression of genes could reflect their functional divergence to some extent. To verify their expression patterns, we analyzed the expression of *CGSs*, *MMTs* and *HMTs* in soybeans (Figs. 5A–5C). In this study, the organs of leaves, stems, flowers and 2w–6w pods were collected and analyzed.

The expression patterns of *GmaCGS1* and *GmaCGS2* were similar. Both of them were highly expressed in leaves and flowers, but significantly decreased during the development process of pods (Fig. 5A). Similarly, the expression models of *GmaMMTs* were analogous, significantly highly expressed in stems, leaves, flowers and 2 week pods, and gradually decreased during the development of pods (Fig. 5B). However, in *GmaHMTs*, the expression patterns were varied (Fig. 5C). For example, the expression of *GmaHMT1* and *GmaHMT3* was significantly higher in the pods and flowers than in

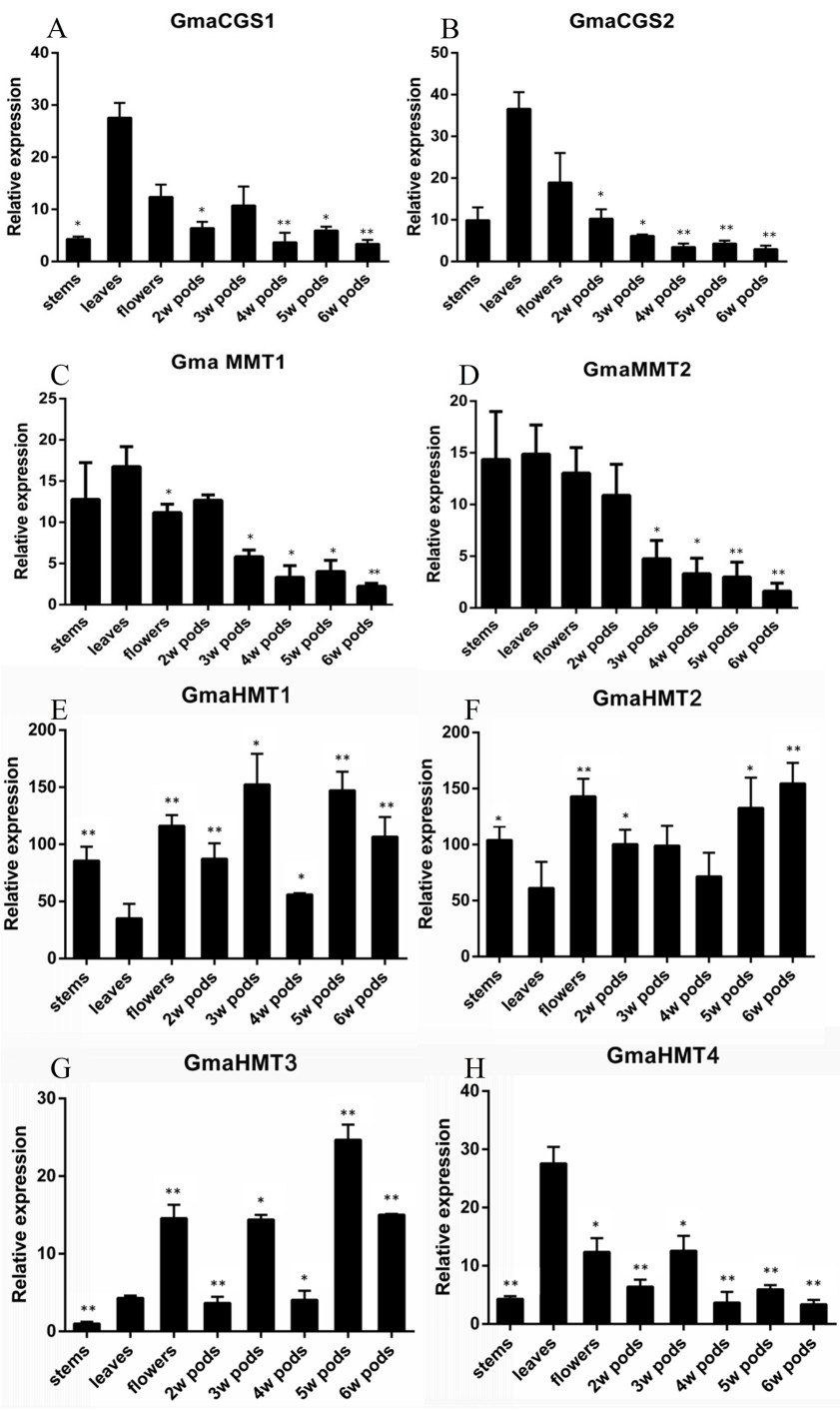

**Figure 5 Expression of the *GmaCGS*, *GmaMMT* and *GmaHMT* genes during soybean development.**
(A–H) The spatio-temporal expression of *GmaCGS1* (A) and *GmaCGS2* (B), *GmaMMT1* (C) and *GmaMMT2* (D), and *GmaHMT1* (E), *GmaHMT2* (F), *GmaHMT3* (G) and *GmaHMT4* (H). The total RNAs were isolated from stems, leaves of 14-day-old seedlings, flowers, and 2-, 4- and 6-week-old pods after fertilization. The *ACTIN* gene was used as an internal control. The experiments were repeated using three independent biological samples. Error bar: standard deviation. The significance was tested in comparison with the expression of each gene in leaves. The * means significance at a $P < 0.05$ level, and the ** means significance at a $P < 0.01$ level.

the leaves and stems. On the contrary, the *GmaHMT2* was fluctuant in different organs, such as leaves, stems, flowers and pods, and the expression levels of *GmaHMT4* were significantly higher in leaves than in flowers, stems and different pods. Above all, the expression patterns within *GmaCGS*s or *GmaMMT*s were consistent, respectively, yet the *GmaHMT*s were distinct from each other. The different expression patterns of the three gene families might be essential to supply Met for the growth and development of soybeans.

## Expression profiles of *CGS*s, *HMT*s and *MMT*s in PLEXdb

To further investigate the gene expression patterns, the tissue expression profiles of *CGS*s, *HMT*s and *MMT*s were widely analyzed in *Arabidopsis*, soybean and rice in the PLEXdb database (Fig. S3A–S3C in File S2). First, the expression patterns of *CGS*s were analyzed. In *Arabidopsis*, both *AthCGS1* and *AthCGS2* were fluctuant in all of the tissues, but their expression levels were generally higher in vegetative tissues than in productive tissues (Fig. S3A in File S2). Nevertheless, the expression intensity of *AthCGS1* (from 11 to 14) and *AthCGS2* (from three to seven) was different. In soybean, only *GmaCGS1* was detected in the database. The expression trends of *GmaCGS1* were similar in the qRT-PCR results, and it was highly expressed in leaves but gradually decreased in seeds and pods (Fig. S3B in File S2). Notably, the expression of *OsaCGS*s was varied. For instance, the expression of *OsaCGS1* was fluctuant in vegetative and productive tissues. The *OsaCGS3* was highly expressed in vegetative tissues, such as leaves, roots and seedlings, while the *OsaCGS5* was higher in endosperms than in vegetative tissues. It was worth noting that their expression intensities were also different, and the highest was found in *OsaCGS3* (intensity from 11 to 14), followed by *OsaCGS1* (intensity from five to eight) and *OsaCGS5* (intensity from one to four).

In *MMT* genes, the expression of *OsaMMT* was not detected (Fig. S4A and S4B in File S2). In *Arabidopsis*, the expression levels of *AthMMT* were basically stable in different tissues, except for seeds. In seeds, the expression level of *AthMMT* was lower than in other vegetative and productive tissues (Fig. S4A in File S2). In soybean, the expression of *GmaMMT1* was high in vegetative tissues and the early stage of seeds, but low in fully grown pods. However, the *GmaMMT2* in different tissues was relative stable compared with *GmaMMT1* (Fig. S4B in File S2). The expression of *HMT*s in the database has also been comprehensively analyzed in our previous article (*Zhao et al., 2018*). Their expression patterns were various, which have been confirmed by the qRT-PCR results in this study. Some *HMT*s were widely expressed in different tissues, while others were particularly highly expressed in specific tissues, such as seeds or leaves. It is worth noting that the expression divergence of *HMT*s was not clade-specific. Generally, the expression of the three key enzymes of CGS, HMT and MMT has experienced varying degrees of divergence.

## Promoter analysis of *CGS*s, *HMT*s and *MMT*s in soybean, *Arabidopsis* and rice

To understand the expression regulation and divergence, the promoters of *CGS*s, *HMT*s and *MMT*s were examined and the *cis*-acting regulatory elements were predicted

*in silico*. A global analysis of regulatory elements in the promoters of *CGS*s, *HMT*s and *MMT*s in soybean, *Arabidopsis* and rice are shown in Table S4 in File S3. In this study, we divided the motifs into two groups: Group 1 (related to levels and locations of expression) and Group 2 (related to responses to stresses) (Table S4 in File S3).

First, the numbers of motifs of *GmaCGS* genes in Group 1 and Group 2 were similar. However, unlike *GmaCGS2*, *GmaCGS1* had two specific motifs, a 5UTR Py-rich stretch and TA-rich region, related to high expression levels, which indicated that the expression levels of *GmaCGS1* might be higher than *GmaCGS2* (Table S4A in File S3). As for *AthCGS*s, the numbers of motifs in the two groups were different. In Group 1, *AthCGS2* (ten motifs) had more motifs than *AthCGS1* (five motifs), but the opposite was the case in Group 2. Considering their similar spatio-temporal expression patterns, the differences in Group 2 might suggest differences in their responses to different stresses (Table S4A in File S3). In rice, the *OsaCGS*s were divided into two classes, *OsaCGS1-4* in Class 2 and *OsaCGS5* in Class 1. *OsaCGS*s in Class 2 (19 in *OsaCGS1*, 24 in *OsaCGS2*, 23 in *OsaCGS2* and 19 in *OsaCGS2*) had more elements responsive to stresses than *OsaCGS5* (six elements), suggesting that the *OsaCGS*s in Class 2 might have an important role in their responses to stresses. In view of expression levels, *OsaCGS2*, *OsaCGS4* and *OsaCGS5* had one 5UTR Py-rich stretch, and *OsaCGS3* had one TA-rich region. In our study, the expression intensity of *OsaCGS3* was higher than that of *OsaCGS1* and *OsaCGS5*, which indicated that the TA-rich region might be necessary to the high expression levels in *OsaCGS*s (Table S4A in File S3). The *MMT*s were relatively conserved, with one or two copies. For example, in rice and *Arabidopsis*, there was only one copy. However, in soybean, there were two copies, and there was a greater number of motifs of *GmaMMT2* (18 motifs in Group 1 and 16 motifs in Group 2) than of *GmaMMT1* (five motifs in Group 1 and eight motifs in Group 2) in the both groups (Table S4B in File S3). Moreover, in *GmaMMT2*, there were 15 enhancers in the promoter, which might be the reason why the expression intensity of *GmaMMT2* was higher than that of *GmaMMT1*.

As for *HMT*s, the *AthHMT*s have been analyzed in our previous study (Table S4C in File S3). In Group 1 and Group 2, the motifs of *AthHMT*s were different. In soybean, there was a greater number of motifs of *GmaHMT4* (24) than of *GmaHMT1–3* in Group 1 (five, seven and one, respectively), while in Group 2, there were fewer motifs of *GmaHMT4* (7) than the others (14, 30 and 15, respectively). Similarly, the motifs of *OsaHMT*s were varied in Group 1 and Group 2 (Table S4C in File S3). Therefore, just as their expression patterns were distinct, their promoters were varied.

## DISCUSSION

### The divergence of *CGS*, *HMT* and *MMT* genes was asynchronous

*CGS*, *MMT* and *HMT* genes are vital to the synthesis of Met in plants (*Datko, Giovanelli & Mudd, 1974*; *Bourgis et al., 1999*; *Ranocha et al., 2001*; *Lee et al., 2008*; *Cohen et al., 2017a*). In this study, their evolutionary histories were reconstructed. Their phylogenetic relationships were different, in which the *MMT*s were conserved during evolution, yet the *CGS* and *HMT* gene families in grasses and seed plants diverged in varying

degrees. Similarly, the gene structures of the *MMT*s and *HMT*s were conserved, but the structure of *CGS*s diverged in the *N*-terminals and intron–exon numbers. Further, the divergence in the *N*-terminals and intron–exon structure in *CGS*s was mainly present in Class 2. Therefore, the evolution of *CGS*, *HMT* and *MMT* gene families was asynchronous.

Although varying degrees of divergence has been detected in *CGS*s, *HMT*s and *MMT*s, they were all under stringent negative selection pressures. The results indicated that the three families did not undergo adaptive evolution. However, a partial subfunctionalization might have occurred. Subfunctionalization in evolution often results from changes in gene expression (*Gallego-Romero, Ruvinsky & Gilad, 2012*; *Wang, Wang & Paterson, 2012*). In our previous results, the subfunctionalization of *HMT*s has occurred in their expression, which might be vital to supplying Met for the development seeds and growth of plants (*Zhao et al., 2018*). However, *MMT*s were similar in their expression patterns. Nevertheless, the expression patterns of *CGS* genes in dicots were also basically consistent. However, in rice, the expression of *CGS* genes was varied. The *OsaCGS*s in Class 2 lost their MTO1 region. In view of the functions of the MTO1 region, which destabilizes the *CGS* mRNA, it seemed that the expression of the *OsaCGS*s might be influenced by the loss of the MTO1 region (*Chiba et al., 1999*). Furthermore, according to the analysis of promoters in *CGS* genes, the *OsaCGS*s without its MTO1 region were rich in the motifs related to stress responses (Table S4A in File S3). The results seemed like that the divergence of *OsaCGS*s in Class 2 might be related to its response to different stresses in rice. However, notably, the similar expression divergence did not occur between *AthCGS*s, although *AthCGS2* has also lost its MTO1 region. Moreover, the numbers of stress response motifs in *AthCGS2* were fewer than in *AthCGS1*. Thus, it seemed that the expression divergence of *CGS*s in grasses and dicots might be independent of the loss of the MTO1 region, or the MTO1 region has a different impact on the grasses and dicots, which need to be further studied. In any case, the *CGS*s, *HMT*s and *MMT*s genes have experienced inconsistent divergence in evolution and expression.

## Evolution pattern of Met biosynthetic pathway in plant lineages

Gene duplication is of huge significance for the evolution of metabolic pathways. The production of gene duplication, two or more copies of genes, leads to the increase of genome size, diversification of enzymes and supplies the raw materials for new properties (*Fondi et al., 2007*; *Lynch & Conery, 2000*; *Zhang, 2003*). Functional innovations in evolution often result from the expressional changes of duplicated genes (*Gallego-Romero, Ruvinsky & Gilad, 2012*; *Wang, Wang & Paterson, 2012*). In our study, the *CGS*, *HMT* and *MMT* genes have been duplicated and diverged during evolution. In addition, previous research has also proposed that the evolution and divergence of metabolic pathways may be disclosed by comparing the sequence and the structure of genes of the same and different routes from organisms (*Fondi et al., 2007*; *Goolsby et al., 2018*).Therefore, combining the evolutionary and expressional pattern of *CGS*, *HMT* and *MMT* genes,

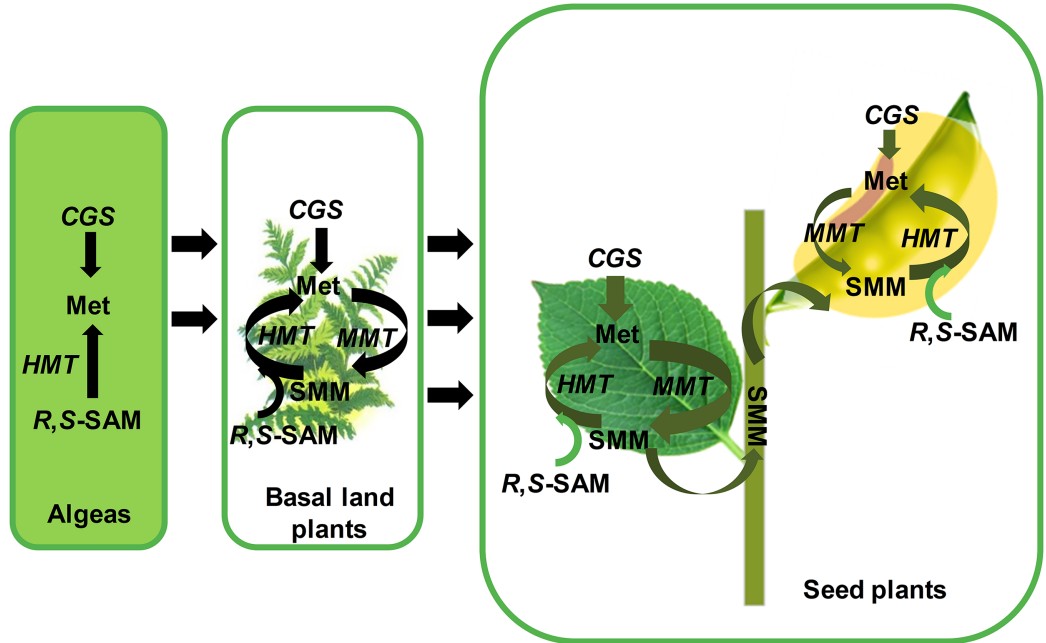

**Figure 6 The functional model of CGSs, MMTs and HMTs to synthesize the methionine in plants.** The enzyme of CGS, MMT and HMT in aspartate family pathway and SMM are italicized. Black arrows indicate the direction of evolution or flux of reaction. The green arrows indicate methionine flux during the growth and seeds development of seed plants. The thickness of green arrows indicates the strength of flow. CGS, cystathionine g-synthase; HMT, homocysteine S-methyltransferases; MMT, met S-methyltransferase; Met, Methionine; *R,S*-SAM, *R,S*-adenosylmethionine; SMM, S-methyl-methionine.

we proposed the evolutionary and functional models of Met biosynthetic pathway in plant lineages (Fig. 6).

In algae, only *CGS* and *HMT* genes were found, which suggested that the Met in algae was only synthesized by the de novo Asp family pathway. As for *HMT* genes, Bradbury has reported that the functions of *HMT* genes contained the ancient, repair of S-adenosylmethionine ((R,S)-AdoMet), and the acquired, SMM catabolism (*Bradbury et al., 2014*). Therefore, in algae, the *HMT* genes might be mainly involved in the repair of (R,S)-AdoMet. Afterward, land plants began to appear in the world. The evolutionary history of *MMTs* suggested that *MMTs* might occur in the ancestor of land plants. In basal land plants, such as moss, *Selaginella moellendorffii* and so on, the *CGSs*, *HMTs* and *MMTs* were grouped together during evolution, respectively, which indicated that the three gene families might not be divergent. Therefore, in basal land plants, Met was supplied to the whole plant by the Asp family pathway and the SMM cycle together. However, due to the loss of expression data in this study, their specific functional patterns were unknown.

In seed plants, different divergence has occurred in the three gene families. First, the *MMT* and *CGS* genes had a high and low expression in vegetative tissues and reproductive tissues, respectively. The high expression of *MMTs* and *CGSs* in vegetative tissues suggested that the Asp family pathway probably supplies the Met largely during the early

vegetative growth of seed plants, which is consistent with Cohen's results (*Cohen et al., 2017b*). It is noteworthy that *HMT* genes have diverged into two clades in seed plants, and their expression has obviously experienced divergence. In the two clades, the *AtHMT1* and *AtHMT3*, and *GmaHMT1-3*s were primarily functioned in seeds, while *AtHMT2* and *GmaHMT4* were largely functioned in leaves or stable in all tissues. The results indicated that more HMTs functioned in seeds in *Arabidopsis* and soybean than the MMTs. Finally, by combining the evolution and expression of *CGS*, *HMT* and *MMT* genes together, we inferred their co-functional models in seed plants as follows (Fig. 5). In vegetative tissues, e.g., leaves, an amount of Met is synthesized, mainly by CGS through the Asp family pathway. Next, a considerable amount of Met enters into the SMM cycle, in which the Met is converted into SMM by MMT. Subsequently, the SMMs are transfered into reproductive tissues (seeds) through phloems (*Bourgis et al., 1999*; *Ranocha et al., 2001*; *Lee et al., 2008*; *Cohen et al., 2017b*). In the meantime, some SMMs are reconverted into the SMM cycle in leaves. In the seeds, the transported SMMs are reconverted into Met by HMTs. This is the main way through which Met is supplied for seed development, especially in the late stage of seed development (*Cohen et al., 2017b*). Furthermore, through the Asp family pathway, the Met is synthesized for seed development (*Cohen et al., 2017b*). It is noteworthy that regardless of the tissues, *R,S*-SAMs are always recovered by HMTs (*Bradbury et al., 2014*). Therefore, adequate Met is supplied for the growth and development of seed plant through the synergistic function of CGSs, HMTs and MMTs.

## CONCLUSIONS

In the present study, the three key enzymes of CGS, MMT and HMT in the biosynthesis of Met were investigated in detail. The evolution patterns of the three gene families have undergone divergence: *MMT*s were conserved, while *CGS*s and *HMT*s have diverged in the grasses or all of the seed plant. The gene structures were conserved, except for *CGS* genes in Class 2. For gene expression, similar to their evolutionary pattern, the *MMT*s were conserved, and the *CGS*s and *HMT*s diverged among tissues. Furthermore, the functions of HMTs were diverged into the repair of (*R,S*)-AdoMet, and SMM catabolism. Therefore, subfunctionalization has occurred in both *CGS* and *HMT* gene families. Finally, based on the evolution and expression divergence of *CGS*s, *HMT*s and *MMT*s, we built the evolution model of Met biosynthetic pathway in plants, which is basically consistent with the evolution of the plants. The model also reveals that *CGS*s, *HMT*s and *MMT*s are essential to supply the Met for the growth and development of different plant lineages.

### Funding

This work was supported by National Natural Science Foundation of China (Grant No. 31600181), Zhejiang Provincial Natural Science Foundation of China (Grant Nos. LQ16C020003 and LY17B020012), and Zhejiang Provincial Major Agriculture Science and Technology Special Sub-project (Grant No. 2016C02050-10-3). The funders had no role in

study design, data collection and analysis, decision to publish, or preparation of the manuscript.

## Grant Disclosures

The following grant information was disclosed by the authors:
National Natural Science Foundation of China: 31600181.
Zhejiang Provincial Natural Science Foundation of China: LQ16C020003 and LY17B020012.
Zhejiang Provincial Major Agriculture Science and Technology Special Sub-project: 2016C02050-10-3.

## Competing Interests

The authors declare that they have no competing interests.

## Author Contributions

- Man Zhao conceived and designed the experiments, analyzed the data, prepared figures and/or tables, authored or reviewed drafts of the paper, approved the final draft.
- Wenyi Wang performed the experiments, analyzed the data, prepared figures and/or tables.
- Lei Wei performed the experiments.
- Peng Chen performed the experiments, contributed reagents/materials/analysis tools.
- Fengjie Yuan contributed reagents/materials/analysis tools, supervision.
- Zhao Wang supervision.
- Xiangxian Ying conceived and designed the experiments, analyzed the data, authored or reviewed drafts of the paper.

## Data Availability

The raw data are provided in the Supplemental Files.

## Supplemental Information

Supplemental information for this article can be found online at http://dx.doi.org/10.7717/peerj.6023#supplemental-information.

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
