# Peer review of "Molecular evolution and expression divergence of three key Met biosynthetic genes in plants: CGS, HMT and MMT"

_PeerJ, doi:10.7717/peerj.6023_

## Round 0.1 · original submission · Major Revisions

I agree with reviewer #2's comments. It is not clearly argued why soybean was emphasized, or why "inconsistent divergence" and gene structure analyses would have anything impact or relevance to cooperation of enzymes for methionine biosynthesis. There is nothing per se wrong with a descriptive paper, but it is unclear why you chose to do this work or what relevance it has.

·

Basic reporting

no comment

Experimental design

no comment

Validity of the findings

Minor comments:
Line 177: “From algaes to angiosperms” should be “From algae to angiosperms”
Line385: The references listed here mentioned the rosette leaves convert Met to SMM, which is then transported to the reproductive tissues to support the synthesis of Met in the developing seeds, but none of them mentioned “most” of the SMMs are transported to the reproductive tissues.
Line387:The reference (Cohen et al., 2017b) could not support your statement: “the transported SMMs are reconverted into Met by the seed-specific expressed HMTs”. AtHMTs are expressed in all plant tissues in Arabidopsis in Cohen et al., 2017.
Table S3B in File S3: what are the red numbers and blue numbers mean?

Additional comments

This study is nicely done but the conclusion can be revised or re-written.
The expression patterns data of HMTs in soybean could not support the conclusion in Fig6” blue HMT were specific expressed in seeds”. If the author want to concluded from the previous paper:” Molecular Evolution and Expression Divergence of HMT Gene Family in Plants” (Zhao et al., 2018), all the data shown in Arabidopsis, soybean, rice, tomato, and Medicago did not help interpret the results” HMT were specific expressed in seeds”.

·

Basic reporting

The article is not clear and unambiguous.

Literature references appear to be appropriate.

The figures and tables are well structured.

There does not appear to be a hypothesis being tested. The paper is entirely descriptive. The summary model shown in Fig 6 does not show more than what has been known from the literature for a decade already.

Experimental design

The work appears to be original.

The research question is not well defined and/or explained.

No justification is given for choosing these specific methionine biosynthesis enzymes, or why other plant methionine biosynthesis enzymes were not included in the analysis.

Analyzing the structure of unrelated genes (e.g. CGS compared with HMT and MMT) is sure to give the result that was found- the genes evolved differently. What is the relevance of such a finding?

Methods are described in sufficient detail.

Validity of the findings

Rationale and benefit to the literature is not clearly stated.

The data is robust and in the case of gene expression analysis, statistically sound and controlled. For most of the data, robustness and statistical analysis is no relevant.

Conclusions are not well stated. Conclusions appear to be trivial.

There is not much speculation that I could find.

Additional comments

The writing, which is in reasonably good English and sentence structure, is full of statements whose meaning is ambiguous and misleading. I present a few examples that highlight the ambiguity, but the following examples are not an exhaustive list and many such instances would need to be re-written. Lines 27-29 states that “CGS, MMT, and HMT genes have been studied, yet how they cooperate with each other to synthesize methionine during evolution in plants is unknown.” It is far from clear what this sentence means. How would the presented gene structure analysis reveal “cooperation with each other”? Also, it isn’t clear what is meant by “methionine biosynthesis during evolution.” Lastly, the sentence ends with “in plants,” but no justification is given for focusing solely on plants. In fact, all three genes evolved prior to the divergence of the kingdoms of life so it isn’t clear what evolutionary answer is being sought by analyzing only the plant lineage. Also, the authors overlook describing how methionine is synthesized in animals, fungi and protists, which is quite relevant given that the enzymes are present in all of these kingdoms of life. In lines 29-31 it is stated that “In the present study, a reconstruction of the evolutionary relationship of CGS, MMT and HMT gene families showed that they inconsistently diverged.” It isn’t clear what “inconsistently diverged” means. Throughout the abstract and text the terms “conserved”, “basically conserved”, “relatively conserved” are used without being defined.
The authors haven’t explained and justified why they analyzed expression of the genes in soybean. Why would soybean provide the evolutionary insight the authors were seeking? They also don’t explain why they chose the selected species to analyze in PLEX. Line 265 “AthCGS1 and AthCGS2 were fluctuant in all of the tissues” suggests that the expression was not consistent, meaning expression pattern showed a great deal of variation. I think what the authors mean is that the expression of the genes differed in different tissues.
Lines 329-330 “Their phylogenetic relationships were distinct,” not clear what this means.
Lines 334-335 “Therefore, the evolution of CGS, HMT and MMT gene families was
335 inconsistent.” What does this mean?
Line 195 “The divergence of genes is reflected in their structures to a certain extent.” What do the authors imply by using the words “to a certain extent”?
In the exon-intron analysis, lines 194-215, the authors analyzed exon-intron number and size. Notably missing is an analysis of intron position in the coding sequence. Intron position is an important aspect of the evolution of genes.

---

## Round 0.2 · Major Revisions

Please address the reviewer concerns, as the current revisions are not yet sufficient.

·

Basic reporting

The authors made a reasonable effort to address reviewer concerns. They didn’t convince me of the significance of their study. The data aside, a large part of the problem is in the writing, which is convoluted and equivocating. I don’t know a way around this other than to advise the authors to hire a writing service.
The abstract needs to be clear and concise because most readers will read only the abstract. So miost of my comments focus on the abstract.
To claim that “The biological functions of CGS, MMT and HMT genes have been studied, respectively, yet both their evolution processes and the evolution pattern of Met biosynthetic pathway in plants are unknown” is a gross overstatement. In fact, quite a great deal of information is known about the evolution of these genes. See doi: 10.1186/s40064-015-1163-8 and doi: 10.1042/BJ20140753, and there are more published studies.
The manuscript fails to mention, particularly so in the abstract, the S-adenosylmethionine repair cycle that plant HMT participates in. The omission is notable because the authors do mention Met biosynthesis and the SMM cycle, which are equally as important as the AdoMet repair cycle. In this regard an important reference is missing on evolution of plant HMT. Bradbury et al. 2014, doi: 10.1042/BJ20140753.
What is meant by “…the divergence of grass and seed plant levels in plants”? Grasses are seed plants.
In the abstract, “…but the CGSs of Class 2 diverged.” Up to this point in the abstract there is no mention of classes of CGS. The authors need to explain what the different classes are, especially in the abstract, because most readers will not know about the classes.
In the abstract, “Nevertheless, all of the genes were under strong negative selections during evolution.” The evidence for this statement needs to be stated. The prior sentence claims that MMTs and HMTs are conserved, but CGSs are not. So what constitutes strong negative selection?
The meaning of the following sentence of the abstract is not decipherable: “The expression patterns of the MMTs were consistent, in which the expression levels in leaves were higher than seeds, but the CGSs and HMTs had diverged in seed plants, which were higher in leaves or seeds, or fluctuated.” I’ll make an effort to rewrite this sentence to reflect the results that are presented. How about this: “The expression of MMT mRNA was consistently higher in leaves than in seeds of Arabidopsis, rice and soybean. Expression of CGS and HMT was not consistently higher in leaves of these three seed plants.” (I have difficulty understanding how the authors decided to generalize their study to “seed plants” when in fact, they analyzed three species, a very small sample of the seed plants, and also they studied on one species, the others were taken from electronic databases.)
The authors should justify how the present work differs from their own recently published paper doi: 10.3390/ijms19041248. This paper seems partly duplicative of the present Peer J manuscript.
The choice of spelling “sulfphur” is not the convention of the International Union of Pure and Applied Chemists, the governing body for chemical nomenclature. The IUPAC convention is “sulfur”, notwithstanding the policy of a few British journals.

Experimental design

No comment

Validity of the findings

No comment

Additional comments

see basic reporting

---

## Round 0.3 · accepted · Accept

Thank you for getting an editing service and for addressing the reviewer comments.